# SPR Sensor-Based Analysis of the Inhibition of Marine Sulfated Glycans on Interactions between Monkeypox Virus Proteins and Glycosaminoglycans

**DOI:** 10.3390/md21050264

**Published:** 2023-04-25

**Authors:** Peng He, Deling Shi, Yunran Li, Ke Xia, Seon Beom Kim, Rohini Dwivedi, Marwa Farrag, Vitor H. Pomin, Robert J. Linhardt, Jonathan S. Dordick, Fuming Zhang

**Affiliations:** 1Center for Biotechnology and Interdisciplinary Studies, Rensselaer Polytechnic Institute, Troy, NY 12180, USA; hep3@rpi.edu (P.H.); shid2@rpi.edu (D.S.); liy77@rpi.edu (Y.L.); xiak@rpi.edu (K.X.); linhar@rpi.edu (R.J.L.); 2Department of Chemistry and Chemical Biology, Rensselaer Polytechnic Institute, Troy, NY 12180, USA; 3Department of Biomedical Engineering, Rensselaer Polytechnic Institute, Troy, NY 12180, USA; 4Department of BioMolecular Sciences, Research Institute of Pharmaceutical Sciences, The University of Mississippi, Oxford, MS 38677, USArdwived1@olemiss.edu (R.D.); mmmoham1@go.olemiss.edu (M.F.); vpomin@olemiss.edu (V.H.P.); 5Department of Food Science & Technology, College of Natural Resources and Life Science, Pusan National University, Miryang 46241, Republic of Korea; 6Departments of Chemical and Biological Engineering, Rensselaer Polytechnic Institute, Troy, NY 12180, USA

**Keywords:** monkeypox virus, protein A29, protein A35, heparin, sea cucumbers, marine sulfated glycans, surface plasmon resonance

## Abstract

Sulfated glycans from marine organisms are excellent sources of naturally occurring glycosaminoglycan (GAG) mimetics that demonstrate therapeutic activities, such as antiviral/microbial infection, anticoagulant, anticancer, and anti-inflammation activities. Many viruses use the heparan sulfate (HS) GAG on the surface of host cells as co-receptors for attachment and initiating cell entry. Therefore, virion–HS interactions have been targeted to develop broad-spectrum antiviral therapeutics. Here we report the potential anti-monkeypox virus (MPXV) activities of eight defined marine sulfated glycans, three fucosylated chondroitin sulfates, and three sulfated fucans extracted from the sea cucumber species *Isostichopus badionotus*, *Holothuria floridana*, and *Pentacta pygmaea*, and the sea urchin *Lytechinus variegatus*, as well as two chemically desulfated derivatives. The inhibitions of these marine sulfated glycans on MPXV A29 and A35 protein–heparin interactions were evaluated using surface plasmon resonance (SPR). These results demonstrated that the viral surface proteins of MPXV A29 and A35 bound to heparin, which is a highly sulfated HS, and sulfated glycans from sea cucumbers showed strong inhibition of MPXV A29 and A35 interactions. The study of molecular interactions between viral proteins and host cell GAGs is important in developing therapeutics for the prevention and treatment of MPXV.

## 1. Introduction

Monkeypox was first found in a colony of cynomolgus monkeys that had smallpox-like symptoms in 1958 [1]. Over the next ten years, several outbreaks among captive monkeys were reported in both the Netherlands and the United States, but no human cases were reported [2]. In 1970 the first human case was diagnosed in a baby boy from the Democratic Republic of Congo (DRC), then the monkeypox virus spread to other African countries and became endemic in Africa [3]. Human monkeypox cases were limited in Africa until 2003, which was when the first monkeypox case was reported in the United States because of the importation of rats from Africa [4]. More imported cases were then sporadically reported in the United Kingdom, Israel, Singapore, and the United States. In May 2022, a monkeypox outbreak in humans started in the UK that then spread across the world. According to the World Health Organization (WHO), by March 2023, 86,391 confirmed cases and 111 deaths were reported, and 110 countries were impacted worldwide [5].

Monkeypox is caused by the monkeypox virus (MPXV), which is a double-stranded DNA virus with a genome of nearly 197 kb and around 190 nonoverlapping ORFs [6]. This zoonotic virus [7] belongs to the family *Poxviridae*, subfamily genus *Orthopoxvirus* (smallpox virus, variola virus, camelpox virus, and cowpox virus all belong to this genus). Monkeypox is genetically divided into two groups: the West African clade (clade I) and the Congo Basin clade (clade II, including clade IIa and clade IIb). The MPXV outbreak nowadays belongs to clade IIb. Unlike the large and quick mutations of single-stranded RNA viruses, such as influenza virus and SARS-CoV-2, the double-stranded DNA structure limited MPXV’s mutations. However, in 2022, MPXV had more than 50 mutations in DNA compared with its 2018 version, and this mutation rate was much higher than before and makes the MPXV a globally concerning infectious disease [8]. Although the smallpox vaccine offers 78% protection against monkeypox [9], there remains a high demand to develop new anti-monkeypox drugs due to the large jump in the viral mutation rate.

The central portion of the genome in different *Orthopoxviruses* is highly conserved, and their proteins are highly homologous. The MPXV has a set of highly conserved *Orthopoxvirus* surface proteins, which can be targeted to block the viral infection [6]. Among these surface proteins, MPXV A29 and M1 are involved in the cellular entry into the intracellular mature virion, while proteins B6 and A35 play important roles in the transmission on the surface of extracellular enveloped virion [10]. The poxvirus A35 protein is reported to be an important virulence factor of poxviruses. A 1000-fold virulence attenuation was detected when the poxvirus lost its A35 proteins. We previously reported that MPXV A29 protein shows high binding affinity to the GAGs, including heparin, chondroitin sulfate (CS), and dermatan sulfate (DS) [11]. We proposed the following three-step model of MPXV host cell entry in which (i) MPXV virions attach to the host cell surface by binding to heparan sulfate proteoglycan (HSPG), (ii) host cell surface proteases initiate viral–host cell membrane fusion, and (iii) virions finally enter the host cell. It is a promising therapeutic target to develop drugs that can bind to A29 to stop the viral entry or binding to the A35 protein, reducing the virulence of the MPXV.

Our previous study found that the sulfated glycans pentosan polysulfate and mucopolysaccharide polysulfate showed strong inhibition of MPXV A29 binding to heparin. Marine-sourced sulfated glycans attracted much attention due to their antiviral activities against the influenza virus, respiratory syncytial virus, cytomegalovirus, and dengue virus. Recently, we reported a series of marine sulfated glycans that showed strong inhibitory activity against SARS-CoV-2 by binding with the viral surface S-protein [12,13,14,15]. In this study, a small library of marine echinoderm sulfated glycans (fucosylated chondroitin sulfates and sulfate fucans; see the structures in Figure 1) derived from three sea cucumbers and one sea urchin were prepared and evaluated for the inhibition activity on the MPXV A29 and A35 protein–heparin interactions using surface plasmon resonance (SPR). Our results demonstrated that MPXV A29 and A35 bind to heparin and the tested sulfated glycans from sea cucumbers showed strong inhibition of the interaction between heparin and A29/A35.

## 2. Results and Discussion

### 2.1. Binding Affinity and Kinetics Measurement of Heparin–MPXV Protein Interactions

Smallpox vaccines based on the vaccinia virus (VACV), such as ACAM2000 and MVA-BN, were shown to be effective in preventing and controlling MPXV by WHO and the CDC [16]. Eight well-studied VACV proteins were applied as immunogenic proteins to neutralize antibodies in humans: A33R, A28L, A27L, A17L, H3, D8L, L1R, and B5R. Among these, MPXV A35 proteins and VACV A33R proteins have a 95.5% genetic similarity [17]. The A35 protein is an extracellular enveloped virus (EEV) that envelops glycoprotein, which is an important component of actin-containing microvilli and plays critical roles in cell-to-cell spread. Heparin is a naturally occurring linear anionic GAG, which is structurally like HS, with a wide distribution in chain molecular weight and sulfation density. Heparin interacts with multiple proteins, endowing heparin with a variety of pharmacological activities, such as anticoagulation, antithrombosis, and anti-inflammation. Many publications demonstrate that many viruses, including SARS-CoV-2, the herpes simplex virus, the influenza virus, and the Zika virus, can infect the host by binding HS on the cell surface [18,19]. HS, as part of the glycocalyx on host cells, facilitates the attachment of viruses to host cells, usually constituting the initial step of viral infection. Heparin and HS also exhibit antiviral activity by blocking viral receptors, thereby inhibiting viruses from engaging with host cells. Recently, three models for how viruses involve HS or therapeutic sulfated GAGs were proposed [20]: (i) viruses use HS as a primary receptor required for infection, such as HSV-1 and HSV-2; (ii) HS serves as a co-receptor that provides localization of the virus on the cell surface by increasing viral concentration and carrying the virus closer to the more specific receptors for viral entry, such as SARS-CoV-2, using HS to facilitate its binding the receptor angiotensin-converting enzyme 2 (ACE2); and (iii) HS can also serve as a protector of the host cell from viral engagement by shielding key receptors. Heparin, which is an HS mimetic, can competitively inhibit the binding of viral proteins to HS on the host cell surface. Here, a heparin chip, as well as CS and DS (GAGs also present on the cell surface) chips, were prepared to evaluate their binding activities with MPXV A35 proteins. Unlike protein A29 binding to the CS and DS [11], neither CS nor DS showed binding to MPXV A35 protein (data not shown). Sensorgrams of interactions of heparin with MPXV A29 and A35 are shown in Figure 2.

Both the binding kinetics and affinity (*ka*, association rate constant; *kd*, dissociation rate constant; and K_D_ (*ka*/*kd*), binding equilibrium dissociation constant) were obtained through globally fitting the entire association and dissociation phases by employing a 1:1 Langmuir binding model. Table 1 shows the kinetic parameters of the interaction between the MPXV A29 and A35 proteins with heparin. The binding affinity of the MPXV A35 protein with heparin was 220 nM and showed a strong binding affinity between heparin and A35 protein. The binding activities of heparin with MPXV A29 protein were also tested as a comparison. The binding kinetic results indicated (i) a similar binding affinity (K_D_) for heparin interaction with A29 and A35: K_D_ = 250 nM for the heparin–A29 interaction and K_D_ = 220 nM for the heparin–A35 interaction; (ii) the A29 protein showed a quick association rate and a quick disassociation rate, while the A35 protein showed a slow association rate and a slow disassociation rate. The interactions between these proteins and heparin were primarily of an electrostatic nature: negatively charged GAGs interacted with positively charged amino acids, including lysine, arginine, and histidine. Based on the sequences of A29 and A35 (Figure 2), some positively charged amino acid clusters served as the heparin-binding domain. 

### 2.2. SPR Solution Competition between Surface-Immobilized Heparin and Marine Sulfated Glycans

The sulfated glycans studied were the sulfated fucan (IbSF) and the fucosylated chondroitin sulfate (IbSFucCS) isolated from the sea cucumber *Isostichopus badionotus*, along with their chemical fully desulfated derivatives desIbSF (polymer of fucose) and desIbFucCS (Figure 1) [21,22]. IbSF has a structure of [→3)-α-Fuc2,4S-(1→3)-α-Fuc2S-(1→3)-α-Fuc2S-(1→3)-α-Fuc-(1→]_n_. IbFucCS has a structure of [→3)-β-GalNAc4,6S-(1→4)β-GlcA-[(3→1)Y]-(1→]_n_, where Y = α-Fuc2,4S (96%) or α-Fuc4S (4%). Both IbSF and IbFucCS showed excellent inhibitory activity against SARS-CoV-2 wild-type and Delta (B.1.617.2) strains by thoroughly disrupting the S-protein interaction with HS on the host cell surface. Fully desulfated IbSF (desIbSF) and IbFucCs (desIbFucCS) were obtained via chemical desulfation using solvolysis after conversion to their pyridinium salt derivatives [23].

Solution/surface competition experiments were performed using SPR to study the ability of *I. badionotus* (Ib)-sourced glycans IbSF, IbFucCS, desIbSF, and desIbFucCS to inhibit the interaction between heparin with MPXV A29 and A35 proteins. The same concentration of Ib glycan (100 µg/mL) was individually mixed in the MPXV A29 or A35 proteins (250 nM). Solution competition studies between heparin and IB glycans are shown in Figure 3. All the Ib glycans inhibited the binding of both the MPXV A29 protein and A35 protein to the surface-immobilized heparin. Soluble heparin inhibited the binding of MPXV A29 and A35 to surface-immobilized heparin by 60% and 72%, respectively. IbSF and IbFucCS showed slightly better results in the inhibitions of A35 binding to surface-immobilized heparin, with 75.8% and 79.9%, respectively. Surprisingly, both IbSF and IbFucCS showed very strong competitive inhibitions, with 95% and 91.5%, respectively. After full desulfation, both desIbSF and desIbFucCS showed a reduced competitive ability to inhibit heparin binding to both the A29 and A35 proteins. This observation indicated that sulfation in the marine sulfated glycans is a key structural element for interactions with MPXV A29 and A35 proteins and, therefore, anti-monkeypox activity.

### 2.3. SPR Solution Competition between Surface-Immobilized Heparin and Holothuria-floridana-Sourced Sulfated Glycans HfSF and HfFucCS

HfSF is a sulfated fucan derived from the sea cucumber *Holothuria floridana* (Hf) with a structure of [→3)-α-Fuc2,4S-(1→3)-α-Fuc-(1→3)-α-Fuc2S-(1→3)-α-Fuc2S-(1→]_n_, while HfFucCS is a fucosylated chondroitin sulfate with a structure of [→3)-β-GalNAc4,6S-(1→4)-β-GlcA-[(3→1)Y]-(1→]_n_, where Y = αFuc2,4S (45%), α-Fuc3,4S (35%), or α-Fuc4S (20%) [24] (Figure 1). These two Hf glycans were found to be good inhibitors of both wild-type and Delta SARS-CoV-2 [13].

Solution/surface competition experiments were performed using SPR to study the ability of HfSF and HfFucCS to inhibit the interactions between heparin with the MPXV A29 and A35 proteins. The same concentration of Hf glycan (100 µg/mL) was individually mixed in the MPXV A29 or A35 proteins (250 nM). Solution competition study results between heparin and Hf glycans are shown in Figure 4. Heparin inhibited the binding of MPXV A29 and A35 binding to surface-immobilized heparin by 60% and 72%, respectively. IbSF and IbFucCS showed slightly better results for the inhibitions of A35 binding to surface-immobilized heparin, with 77.4% and 77%, respectively. However, IbSF and IbFucCS showed strong competitive inhibition results, with 92.4% and 86.1%, respectively.

### 2.4. SPR Solution Competition between Surface-Immobilized Heparin and two Marine-Soured Sulfated Glycans LvSF and PpFucCS

LvSF is a sulfated fucan isolated from the sea urchin *Lytechinus variegatus* and has a structure of [→3)-α-Fuc2,4S-(1→3)-αFuc2S-(1→3)-α-Fuc2S-(1→3)-α-Fuc4S-(1→]_n_ [14], while PpFucCS is the fucosylated chondroitin sulfate isolated from the sea cucumber *Pentacta pygmaea* and has a structure of [→3)-β-GalNAcX(1→4)-β-GlcA-[(3→1)Y]-(1→]_n_, where X = 4S (80%), 6S (10%), or non-sulfated (10%), and Y = α-Fuc2,4S (40%), αFuc2,4S-(1→4)-α-Fuc (30%), or α-Fuc4S (30%) [15] (Figure 1).

Solution/surface competition experiments were performed using SPR to study the inhibition of LvSF and PpFucCS for the interaction between heparin with MPXV A29 and A35 proteins. The same concentration of Hf glycan (100 µg/mL) was individually mixed in the MPXV A29 or A35 proteins (250 nM). The solution competition study results between heparin and sulfated glycans are shown in Figure 5. Heparin inhibited the binding of MPXV A29 and A35 to surface-immobilized heparin by 60% and 72%, respectively. LvSF and PpFucCS showed a better result of the inhibition of A35′s binding to surface-immobilized heparin, with 83.1% and 89%, respectively. LvSF and PpSFucCS showed strong competitive inhibition results, with 76.4% and 87%, respectively.

All six natural marine-derived sulfated glycans (IbSF, IbFucCS, HfSF, HfFucCS, PpFucCS, LvSF) showed the ability to inhibit the interactions between monkeypox virus proteins (both A29 and A35) and surface-immobilized heparin. However, both desulfated glycans, desIbSF and desIbFucCS showed significantly reduced binding properties of both viral proteins to surface-immobilized heparin (Table 2). This dataset indicates that sulfation is a key structural element for the inhibition activity of marine sulfated glycans. All six natural marine sulfated glycans exhibited remarkable inhibition activity against surface-immobilized heparin binding with the MPXV A29 protein. Among the three kinds of fucosylated chondroitin sulfates, IbFucCS had the highest sulfation level (96% branching disulfated fucoses) and exhibited the best inhibitory property. HfFucCS (80% branching disulfated fucoses) and PpFucCS (70% branching disulfated fucoses) showed similar inhibitory activity despite the lower sulfation content in PpFucCS. Although IbSF, LvSF, and HfSF are all tetrasaccharide-repeating sulfated fucans, IbSF and HfSF are tetrasulfated per tetrasaccharide building blocks, while LvSF is pentasulfated per tetrasaccharide building blocks. IbSF and HfSF showed very similar inhibitory activities and, interestingly, higher action than LvSF despite the higher sulfation content of the latter. This indicates that the sulfation pattern can play a more significant role in the interactions with the monkeypox proteins than the sulfation content. Although all tested sulfated glycans showed strong inhibitory activity against both viral proteins in the SPR-based binding to the heparin surface, no clear correlations between the structural features of these glycans and binding properties could be drawn. Similar results were observed in our previous study on the inhibitory activity of these sulfated glycans against evolving SARS-CoV-2 strains [13].

## 3. Materials and Methods

### 3.1. Materials

Eight marine sulfated glycans (IbSF, desIbSF, IbFucCS, desIbFucCS, PpFucCS, LvSF, HfSF, HfFucCS) from the sea cucumbers *I. badionotus* and *P. pygmaea*, sea urchin *L. variegatus*, and the Florida sea cucumber *Holothuria floridana* were purified [19,21] in Dr. Pomin’s lab at the University of Mississippi. MPXV A29 and A35 proteins were purchased from Sino Biological Inc. (Wayne, PA, USA). The recombinant MPXV protein A29 consists of 97 amino acids and has a predicted molecular mass of 11.36 kDa. The recombinant MPXV protein A35 consists of 135 amino acids and has a predicted molecular mass of 15.12 kDa (see the amino acid sequences in Figure 2). Porcine intestinal heparin was purchased from Celsus Laboratories (Cincinnati, OH, USA) with an average molecular weight of 15 kDa and polydispersity of 1.4. Sensor SA chips were purchased from Cytiva (Uppsala, Sweden). SPR measurements were performed on a BIAcore 3000 or T200 SPR (Uppsala, Sweden), which was operated using Biaevaluation software (version 4.0.1 or 3.2).

### 3.2. Preparation of Heparin Biochips

The biotinylated heparin was prepared using the following method: 1 mg of heparin and 1 mg of amine-PEG_3_-Biotin (Thermo Scientific, Waltham, MA, USA) were dissolved in 200 µL water, and then 5 mg NaCNBH_3_ was added. The mixture was incubated at 70 ℃ for 24 h, then another 5 mg NaCNBH_3_ was added, and the reaction was incubated for another 24 h. After completing the reaction, the mixture was desalted with a spin column (3000 molecular weight cut-off). Biotinylated heparin was freeze-dried for chip preparation. A heparin SA chip for the SPR study was made using the following protocol: 20 µL solution of the biotinylated heparin (0.1 mg/mL) in HBS-EP^+^ buffer was injected over flow cells 2 to 4 of the SA chip at a flow rate of 10 μL/min. Furthermore, flow cell 1 was immobilized using biotin as a reference channel using the same method.

### 3.3. Binding Kinetics and Affinity Studies of the Interaction between Heparin and the MPXV A35 Protein

The MPXV A35 protein was diluted with HBS-EP^+^ buffer (pH 7.4). Different dilutions of A35 protein were injected at a flow rate of 30 µL/min. At the end of each injection, the same buffer was allowed to flow over the sensor surface to facilitate dissociation for 180 s. The SPR chip was regenerated by injecting it with 30 µL of 2 M NaCl. The response was monitored using a sensorgram at 25 °C.

### 3.4. Inhibition Activity of the Marine Sulfated Glycans on Heparin–MPXV Protein Interactions

To evaluate the inhibition of the MPXV protein–heparin interaction, 250 nM of MPXV protein was premixed with 100 µg/mL of different glycans in HBS-EP^+^ buffer (pH 7.4) and injected over the heparin chip with a flow rate of 30 µL/min. The same buffer was allowed to flow over the sensor surface to facilitate dissociation after each injection. A 30 µL injection of 2 M NaCl was used for the regeneration of the sensor surface. Sensorgrams were monitored at 25 ℃. MPXV proteins were used in the control experiments to make sure the surface was completely regenerated. When the binding sites of MPXV proteins were occupied by our glycan samples, the binding of premixed proteins with glycan samples to the surface-immobilized heparin was decreased with an RU attenuation of the SPR.

## 4. Conclusions

The current study demonstrated that MPXV A29 and A35 were strongly bound to heparin. Solution competition analysis between surface-immobilized heparin with eight defined marine sulfated glycans (IbSF, desIbSF, IbFucCS, desIbFucCS, PpFucCS, LvSF, HfSF, HfFucCS) from sea cucumber was performed by using an SPR competition assay. These results showed that the natural marine sulfated glycans (IbSF, IbFucCS, PpFucCS, LvSF, HfSF, HfFucCS) provided outstanding inhibitory activity of chip-surface heparin binding to the MPXV A29 and A35 proteins. However, the fully desulfated IbSF (desIbSF) and fully desulfated desIbFucCS showed only a weak inhibition activity. Our findings demonstrate that some sulfated glycans from cucumbers are promising natural MPXV inhibitors through binding with two critical MPXV proteins, namely, A29 and A35. The study of molecular interactions should facilitate the development of therapeutics for the prevention and treatment of MPXV. Additional studies will be needed in the future to uncover the potential in vitro and in vivo properties of marine sulfated glycans against monkeypox infection.

## Figures and Tables

**Figure 1 marinedrugs-21-00264-f001:**
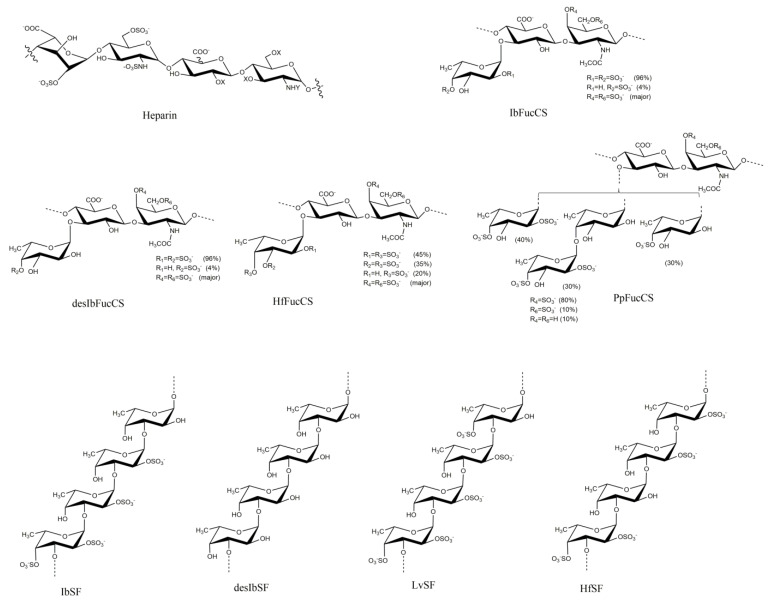
Chemical structures of heparin and marine sulfated glycans.

**Figure 2 marinedrugs-21-00264-f002:**
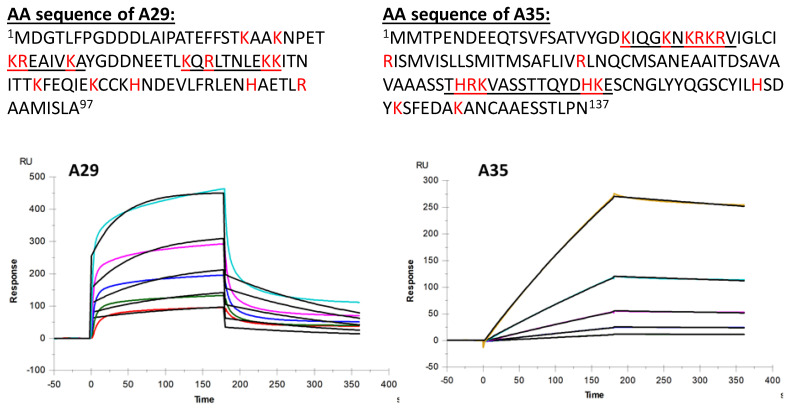
Amino acid sequence of MPXV A29 and A35 proteins (positively charged amino acids histidine (H), lysine (K), and arginine (R) are in red) and SPR sensorgrams of MPXV A29 and A35 protein binding with heparin. Concentrations of A35 protein were 1000, 500, 250, 125, 62.5 and 31.3 nM (from top to bottom, respectively).

**Figure 3 marinedrugs-21-00264-f003:**
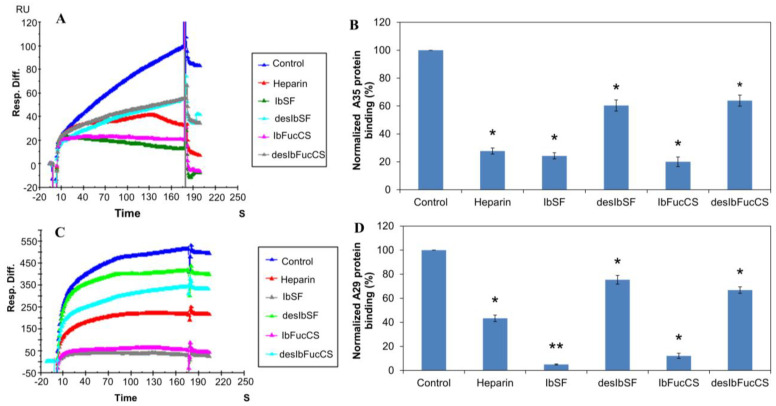
Solution competition between heparin and heparin Ib glycans. (**A**) SPR sensorgrams of the MPXV A35 protein–heparin interaction competing with different Ib glycans. The concentration of the MPXV A35 protein was 250 nM mixed with 100 µg/mL of different Ib glycans. (**B**) Bar graphs (based on triplicate experiments with standard deviation) of normalized MPXV A35 protein binding preference to surface heparin by competing with different Ib glycans. (**C**) SPR sensorgrams of the MPXV A29 protein–heparin interaction competing with different Ib glycans. The concentration of the MPXV A29 protein was 250 nM mixed with 100 µg/mL of different Ib glycans. (**D**) Bar graphs (based on triplicate experiments with standard deviations) of the normalized MPXV A29 protein binding preference to surface heparin by competing with different Ib glycans. Statistical analysis was performed using an unpaired two-tailed *t*-test (*: *p* ≤ 0.05 compared with the control, **: *p* ≤ 0.01 compared with the control).

**Figure 4 marinedrugs-21-00264-f004:**
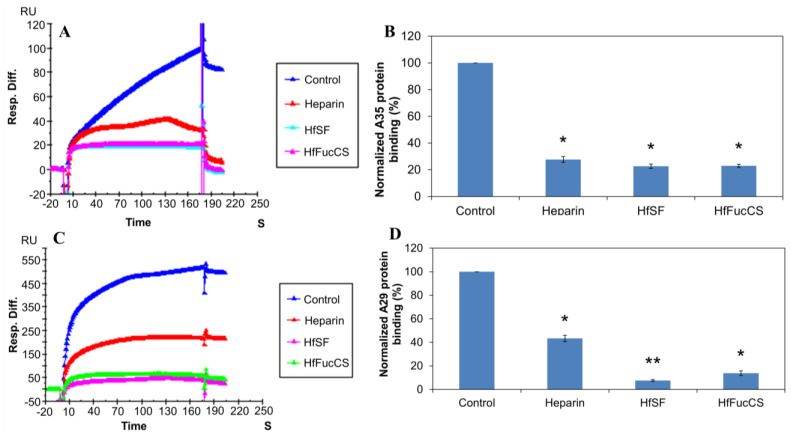
Solution competition between heparin and Hf glycans. (**A**) SPR sensorgrams of the MPXV A35 protein–heparin interaction competing with different Hf glycans. The concentration of the MPXV A35 protein was 250 nM mixed with 100 µg/mL of different Hf glycans. (**B**) Bar graphs (based on triplicate experiments with standard deviations) of the normalized MPXV A35 protein binding preference to surface heparin by competing with different Hf glycans. (**C**) SPR sensorgrams of the MPXV A29 protein–heparin interaction competing with different Hf glycans. The concentration of the MPXV A29 protein was 250 nM mixed with 100 µg/mL of different Hf glycans. (**D**) Bar graphs (based on triplicate experiments with standard deviations) of the normalized MPXV A29 protein binding preference to surface heparin by competing with different Hf glycans. Statistical analysis was performed using an unpaired two-tailed *t*-test (*: *p* ≤ 0.05 compared with the control, **: *p* ≤ 0.01 compared with the control).

**Figure 5 marinedrugs-21-00264-f005:**
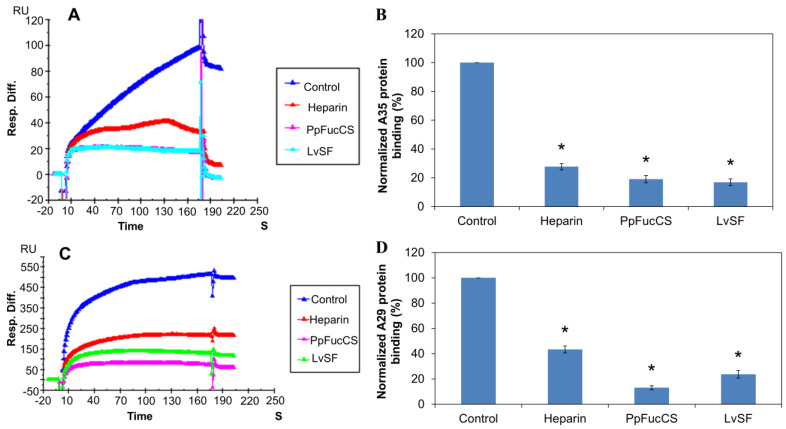
Solution competition between heparin and LvSF and PpFucCS glycans. (**A**) SPR sensorgrams of the MPXV A35 protein–heparin interaction competition with different LvSF and PpFucCS glycans. The concentration of the MPXV A35 protein was 250 nM mixed with 100 µg/mL of different LvSF and PpFucCS glycans. (**B**) Bar graphs (based on triplicate experiments with standard deviations) of the normalized MPXV A35 protein binding preference to surface heparin by competing with different LvSF and PpFucCS glycans. (**C**) SPR sensorgrams of the MPXV A29 protein–heparin interaction competing with different LvSF and PpFucCS glycans. The concentration of the MPXV A29 protein was 250 nM mixed with 100 µg/mL of different LvSF and PpFucCS glycans. (**D**) Bar graphs (based on triplicate experiments with standard deviations) of the normalized MPXV A29 protein binding preference to surface heparin by competing with different LvSF and PpFucCS glycans. Statistical analysis was performed using an unpaired two-tailed *t*-test (*: *p* ≤ 0.05 compared with the control).

**Table 1 marinedrugs-21-00264-t001:** Kinetic data of the interaction of the MPXV A29 and A35 proteins with heparin.

	*k* (M^−1^S^−1^)	*K_d_* (S^−1^)	K_D_ (M)
A29 protein	2.0 × 10^4^ (±190) *	5.1 × 10^−3^ (±4.1 × 10^−5^) *	2.5 × 10^−7^
A35 protein	1.8 × 10^3^ (±14) *	4.0 × 10^−4^ (±1.6 × 10^−6^) *	2.2 × 10^−7^

* The data with (±) in parentheses are the standard deviations (SDs) from the global fitting of five injections.

**Table 2 marinedrugs-21-00264-t002:** Summary of solution competition between heparin and eight marine-derived glycans binding to MPXV proteins ^a^.

	Control ^b^	Heparin	IbSF	desIbSF	IbFucCS	desIbFucCS	HfSF	HfFucCS	PpFucCS	LvSF
Normalized A29 protein binding (%)	100	43.3 *	5.0 **	75.4 *	7.6 *	66.8 *	8.5 **	13.9 *	13.0 *	23.6 *
Normalized A35 protein binding (%)	100	27.7 *	24.3 *	60.4 *	20.1 *	63.9 *	23.9 *	23.0 *	19.0 *	16.9 *

^a^ SPR sensorgrams of MPXV A29/A35 protein–heparin interaction competing with eight different marine-derived glycans. The concentration of proteins was 250 nM mixed with 100 µg/mL of different glycans. ^b^ The control had the same concentration of A29/A35 protein (125 nM) without any mixed glycans. Statistical analysis was performed using an unpaired two-tailed t-test (*: *p* ≤ 0.05 compared with the control, **: *p* ≤ 0.01 compared with the control).

## Data Availability

Data presented in this study are available on request.

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
