# Peer review of "SPR Sensor-Based Analysis of the Inhibition of Marine Sulfated Glycans on Interactions between Monkeypox Virus Proteins and Glycosaminoglycans"

_marinedrugs, 2023, doi:10.3390/md21050264_

Round 1
Reviewer 1 Report
This work, continues and deepens the study begun in Shi, D.; He, P.; Song, Y.; Cheng, S.; Linhardt, R.J.; Dordick, J.S.; Chi, L.; Zhang, F., Kinetic and structural aspects of glycosamino-301 glycan-monkeypox virus protein A29 interactions using surface plasmon resonance. Molecules 2022, 27, (18), 5898 and it evaluates the disruptive effects of the interaction of two monkeypox virus proteins with their cellular receptor of a series of sulfated glycans from marine organisms.
The work is well written, the experiments are well planned and the results allow us to reach the conclusions that are detailed.
I consider that the evaluation of the antiviral effect of the compounds in infections with the virus is necessary, at least in vitro, to corroborate that the compounds have an inhibitory effect on the multiplication of MPXV and can be considered as a potential therapeutic treatment.
In virology, family and genera names should be written in italics, for example in line 46.
Line 229 missing a space, 1 mg for 1mg.
Author Response
This work, continues and deepens the study begun in Shi, D.; He, P.; Song, Y.; Cheng, S.; Linhardt, R.J.; Dordick, J.S.; Chi, L.; Zhang, F., Kinetic and structural aspects of glycosamino-301 glycan-monkeypox virus protein A29 interactions using surface plasmon resonance. Molecules 2022, 27, (18), 5898 and it evaluates the disruptive effects of the interaction of two monkeypox virus proteins with their cellular receptor of a series of sulfated glycans from marine organisms. The work is well written, the experiments are well planned and the results allow us to reach the conclusions that are detailed.
Response: Thanks for the positive and constructive comments on our manuscript.
I consider that the evaluation of the antiviral effect of the compounds in infections with the virus is necessary, at least in vitro, to corroborate that the compounds have an inhibitory effect on the multiplication of MPXV and can be considered as a potential therapeutic treatment.
Response: In this study we focused on the evaluation of 8 marine-derived sulfated glycans’ inhibition activity on two MPVX proteins, A29 (involved in cellular entry on intracellular mature virion) and A35 (involved in the transmission on the surface of extracellular enveloped virion). We believe this is a very useful library of sulfated polysaccharides of unique structures and activity against monkeypox has never been published before. This library of sugars is exclusive to our group. This is an innovation in our work currently submitted to Marine Drugs and reliable mechanistic data have been assembled here for a publication. We agree that in vitro and in vivo assays are fundamental for drug development of these marine sugars as antiviral. We are adding these in vitro and in vivo assays in a grant proposal and upon funding, we will surely expand our studies in that direction. Data collected from future in vitro and in vivo assays will be published in another manuscript. Thanks for your comment on this matter.
In virology, family and genera names should be written in italics, for example in line 46.
Response: They have been corrected as suggested.
Line 229 missing a space, 1 mg for 1mg.
Response: It has been corrected as suggested.
Reviewer 2 Report
Journal-Marine Drugs (ISSN 1660-3397)
Title- Marine Sulfated Glycans Inhibit Interactions between Monkeypox Virus Protein and Glycosaminoglycans
Special Issue- Advances in Oligosaccharides and Polysaccharide Modifications in Marine Bioresources
The article ‘Marine Sulfated Glycans Inhibit Interactions between Monkeypox Virus Protein and Glycosaminoglycans’ by He et al is a relevant topic of current interest, where more studies on the antiviral marine polysaccharides are explored worldwide. In this study, authors evaluated the antiviral interactions of eight marine sulfated glycans, three fucosylated chon-droitin sulfates and three sulfated fucans extracted against monkeypox virus (MPXV). The glycans were obtained from various sea cucumber species Isostichopus badionotus, Holothuria floridana, and Pentacta pygmaea, also from the sea urchin Lytechinus variegatus, and two chemically desulfated derivatives. SPR based sensor analysis were done to analyze the MPXV A29 and A35 protein-heparin interactions to understand the inhibitory effects of glycans. In a therapeutic point of view, this sensor-based study could give some inputs to identify the potential glycans which could be used for further antiviral evaluations. The article is interesting, however, to improve the quality of the manuscript some changes could be addressed before considering it for publication in its current form.
1. HS is abbreviated in abstract, and in line number 97, HS is listed as abbreviation and later on in line number 102, ‘heparan sulfate (HS)’ the full form with abbreviation is listed please follow a standard system.
2. Figure 2 needs to be well explained for the reader to understand the mechanism, SPR sensogram mechanism, and relevance, various concentrations effect on the protein binding of both A29 and A35. The current explanation is very limited.
3. Figure quality needs to be improved, try to include high quality figures, in the current figures the characters are also not clear.
4. In figure 3 both IbSF and IbFucCS showed excellent inhibitory effects compared to the other competing proteins. Please explain the possible mechanism or interaction behind the activity. If it is sulfation degree, what will be the interaction in detail? Please include more discussion.
5. Have you done a comparative analysis of the 8 glycans together with the sensor, heparin and control? Probably that would have been an easier comparison to evaluate the highest inhibitory effect.
A table comparing all the inhibitory effects would be easier to identify and compare the 8 glycans together in the discussion.
6. Other than SPR analysis, have you done any other characterization to compare the antiviral effect of this gylcans?
7. What is the stability of these glycans in a physiological solution and different pH? Was any stability analysis performed?
8. What was the pH conditions during the SPR sensor analysis?
9. Have you done or used any statistical analysis in this study? If so, please include the details. 10. Title seems very general. Probably it would be a great idea to reflect the main aspect of the study as sensor-based analysis in the title as well, like ‘SPR sensor based analysis of’.

Author Response
The article ‘Marine Sulfated Glycans Inhibit Interactions between Monkeypox Virus Protein and Glycosaminoglycans’ by He et al is a relevant topic of current interest, where more studies on the antiviral marine polysaccharides are explored worldwide. In this study, authors evaluated the antiviral interactions of eight marine sulfated glycans, three fucosylated chon-droitin sulfates and three sulfated fucans extracted against monkeypox virus (MPXV). The glycans were obtained from various sea cucumber species Isostichopus badionotus, Holothuria floridana, and Pentacta pygmaea, also from the sea urchin Lytechinus variegatus, and two chemically desulfated derivatives. SPR based sensor analysis were done to analyze the MPXV A29 and A35 protein-heparin interactions to understand the inhibitory effects of glycans. In a therapeutic point of view, this sensor-based study could give some inputs to identify the potential glycans which could be used for further antiviral evaluations. The article is interesting, however, to improve the quality of the manuscript some changes could be addressed before considering it for publication in its current form.
Response: Thanks for these encouraging and constructive comments on our manuscript.
- HS is abbreviated in abstract, and in line number 97, HS is listed as abbreviation and later on in line number 102, ‘heparan sulfate (HS)’ the full form with abbreviation is listed please follow a standard system.
Response: It has been corrected as suggested.
- Figure 2 needs to be well explained for the reader to understand the mechanism, SPR sensogram mechanism, and relevance, various concentrations effect on the protein binding of both A29 and A35. The current explanation is very limited.
Response: The major mechanism of the interactions between these protein and heparin is based on electrostatic: negatively charged GAGs interacting with positive charged amino acid residues including lysine, arginine, and histidine. Based on the sequences of A29 and A35 (Figure 2), there are a few positive charged amino acid residues clusters, which can be served as heparin-binding domains. The sensorgrams from both protein-heparin analysis shows concentration dependent profile for the interaction: higher concentration of protein injected, higher binding signal (RU) observed. Both sets of sensorgrams fit well with 1:1 Langmuir-binding model. We have added more explanation regarding this matter in the revised paper (see pages 4 and 5, lines 137-141).
- Figure quality needs to be improved, try to include high quality figures, in the current figures the characters are also not clear.
Response: We have included therein figures in high resolution.
- In figure 3 both IbSF and IbFucCS showed excellent inhibitory effects compared to the other competing proteins. Please explain the possible mechanism or interaction behind the activity. If it is sulfation degree, what will be the interaction in detail? Please include more discussion.
Response: Compared with IbSF and IbFucCS, desulfated version of these two molecules showed reduced the inhibition on heparin-MPXV protein interaction. This observation indicates that sulfation in the marine sulfated glycans are a key structural feature for interaction with MPXV A29 and A35 proteins and therefore, anti-monkeypox activity. Based on the inhibition results: Fig. 3 (A and B) – A35, IbSF 24.3%, IbFucCS 20.1%, (C and D) – A29, IbSF ~5%, IbFucCS 7.6%, Fig4. (A and B) – A35,. HfSF 23.9%, HfFucCS 23.0%, (C and D) – A29, HfSF 8.5%, HfFucCS 13.9%, Fig. 5 (A and B) – A35, PpFucCS 19.0%, LvSF 16.9%, (C and D) – A29, PpFucCS 13.0%, LvSF 23.6%, all tested sulfated sultated glycans showed strong inhibitory activity against both proteins binding to heparin, but no correlations between glycan structural features and binding inhibition could be drawn.
- 5. Have you done a comparative analysis of the 8 glycans together with the sensor, heparin and control? Probably that would have been an easier comparison to evaluate the highest inhibitory effect. A table comparing all the inhibitory effects would be easier to identify and compare the 8 glycans together in the discussion.
Response: A summary table (Table 2) for comparative analysis of the 8 marine glycans together with heparin and control has been provided in our revised manuscript.
- 6. Other than SPR analysis, have you done any other characterization to compare the antiviral effect of this gylcans?
Response: In this study we focused on the evaluation of 8 marine-derived sulfated glycans’ inhibition activity on two MPVX proteins, A29 (involved in cellular entry on intracellular mature virion) and A35 (involved in the transmission on the surface of extracellular enveloped virion). We believe this is a very useful library of sulfated polysaccharides of unique structures and activity against monkeypox has never been published before. This library of sugars is exclusive to our group. This is an innovation in our work currently submitted to Marine Drugs and reliable mechanistic data have been assembled here for a publication. We agree that in vitro and in vivo assays are fundamental for drug development of these marine sugars as antiviral. We are adding these in vitro and in vivo assays in a grant proposal and upon funding, we will surely expand our studies in that direction. Data collected from future in vitro and in vivo assays will be published in another manuscript. Thanks for your comment on this matter.
- What is the stability of these glycans in a physiological solution and different pH? Was any stability analysis performed?
Response: Like glycosaminoglycans and unlike proteins and enzymes, marine sulfated glycans are stable in a large range of physicochemical conditions such as pH, temperature and ionic strength. The stability of these compounds is well known. In physiological conditions, they are usually at their sodiated forms. These marine sulfated glycans have been handled by Pomin’s group for many years and many publications have appeared in the literature regarding these molecules. See Pomin’s publication record regarding these marine sugars.
- What was the pH conditions during the SPR sensor analysis?
Response: The pH condition is 7.4 for all the SPR analysis and it is added to the manuscript.
- Have you done or used any statistical analysis in this study? If so, please include the details.
Response: Statistical analysis data has been provided in our revised manuscript.
- Title seems very general. Probably it would be a great idea to reflect the main aspect of the study as sensor-based analysis in the title as well, like ‘SPR sensor based analysis of’.
Response: Thanks for your suggestion. The title has been revised as “SPR Sensor Based Analysis of the Inhibition of Marine Sulfated Glycans on Interactions between Monkeypox Virus Protein and Glycosaminoglycans”
Round 2
Reviewer 1 Report
Following the authors' response, I agree to the publication of this latest version of the manuscript.